# A New Kind of Absolute Magnetic Encoder

**DOI:** 10.3390/s21093095

**Published:** 2021-04-29

**Authors:** Tong Feng, Wenlu Chen, Jinji Qiu, Shuanghui Hao

**Affiliations:** School of Mechatronics Engineering, Harbin Institute of Technology, Harbin 150001, China; ariahoshi.tf@gmail.com (T.F.); chenwenlu2021@163.com (W.C.); 18846446691@163.com (J.Q.)

**Keywords:** absolute angular displacement sensor, Hall-effect sensor, combined magnetic encoder, multi-pole alnico magnet

## Abstract

To fulfill the requirement of fields such as robotics, aviation, and special machining, motors with quill shafts or outer rotors have been used. For these special motors, the photoelectric encoder’s volume is normally too big and easy to be polluted by oil or dust; magnetic encoder normally has poor accuracy, and alnico piece may not provide enough magnetic field coverage area. The aim of this essay is to find a new structure of magnetic encoder to improve the precision and magnetic field coverage area. By using two multi-pole alnico rings with a different number of pole pairs to provide a magnetic field, the coverage area could be improved. The position differences between two alnicos pole positions are used to calculate absolute angle value, so the accuracy of the encoder could be absolute and no less than that of a combined magnetic encoder with the same number of pole pairs. A special algorithm is proposed for decoding. This new kind of magnetic encoder could be used on special motors with quill shafts or outer rotors. Its volume and weight are less than the photoelectric encoder and have better performance on antipollution. The alnico ring is easy to modify to suit the structure of the motor.

## 1. Introduction

Servo motor systems are widely used nowadays. In fields such as robotics, aviation, and special machining, the system needs to be miniaturized and lightweight owing to the requirements of the working environment. Motors with quill shafts or outer rotors have been proposed and used to meet these needs. To fulfill the required control accuracy, a suitable angular displacement sensor needs to be used in the system. Two kinds of sensors commonly used in servo motor systems are the photoelectric encoder and magnetic encoder. A photoelectric encoder has high accuracy but normally has a relatively big size and high cost and performs poorly in an environment with vibration disturbance or low cleanliness. A magnetic encoder is cheaper and smaller and is strongly resistant to pollution and disturbance, but its accuracy is poorer.

Research on photoelectric encoders has mainly focused on acclimatization and reducing volume. Zhao transformed the signal current from the encoder′s receiver into a signal voltage and then processed the signal through other chips, which reduced the effects of environmental temperature and working time on the light source and receiver [1,2]. To reduce the volume, Wang and Xi used single-ring coding to engrave the code disk of the photoelectric encoder, making the code disk smaller [3,4]. Wang also created a metal code disk to improve the impact resistance of the encoder [3]. Yu used an image detector instead with a traditional moiré fringe method and designed a small photographic encoder with a diameter of 50 mm [5].

For a magnetic encoder, the research emphasis has mainly been on increasing the encoder’s accuracy and resolution and reducing the effects of disturbance. The method of the look-up table is often used in early research. Nakano used CMOS magnetic sensors to detect magnetic fields, then created a table of sensors offset and used it to correct the final output [6]. Hao created a table of the relationship between Hall sensors’ output and photoelectric encoder’s output for calibration, increased the accuracy of the encoder to 13-bits [7]. This method is still used in recent research. Park used a gear system to combine the main shaft with some subshafts, and the signals from alnicos on subshafts could be broken to subdivide the signal from the alnico of the main shaft [8]. Then Nguyen proposed a self-referencing lookup-table algorithm to improve the accuracy of this encoder [9]. Another method is improving the structure of the encoder. Pavel designed an encoder with a Hall ring and a specially designed permanent magnet to achieve ± 0.1 degrees without calibration [10]. Zhang used coils stimulated by the external current to generate the magnetic field, then used two groups of secondary windings to detect the change of magnetic field and calculate angular by signal processing circuit [11]. Yamamoto designed a magnetic encoder with an eccentric structure, using four linear Hall sensors to receive signals from the eccentrically rotating multi-pole alnico and then calculate the absolute angle through a least-squares algorithm [12]. Wang built a relationship between the time pulse and magnetic signal from magnetoresistance sensors and then calculated the angle by counting the time pulse [13]. Tran used a combined encoder and designed an algorithm based on an adaptive linear neural network to reduce the effects of noise and offset [14].

A motor with a quill shaft always has a relatively large radial dimension. A photoelectric encoder needs more grating to be processed and has a higher cost. From the studies above, it is evident that the oil and dust pollution of the light source and receiver still has not been solved. For a magnetic encoder, the magnetic field from the normal alnico has a saddle-type distortion and limited coverage area. To cover the end face of a quill shaft motor, the alnico needs to be larger but will be easy to break owing to stress. Ripka used a coil as a magnetic field source with a working distance of 20 m, but the accuracy was poor [15]. Therefore, finding a suitable encoder for a motor with a quill shaft is an important problem to research.

This paper proposes a new kind of magnetic encoder and a method to determine its absolute angle. This encoder uses alnico rings instead of permanent magnet sheets, so it is more suitable for a motor with a quill shaft than other magnet encoders. By using the position relationship between outer and inner alnico rings to calculate the angular value, the method could eliminate the effects of processing and installing. The structure of this encoder is flexible, and the algorithm is simple.

## 2. Principle of Angle Measurement

Figure 1 shows the schematic of the magnetic encoder’s signal generation principle. Two multi-pole alnico rings are installed on a motor’s rotary shaft coaxially and rotate with it, outputting a magnetic signal that correlates with angle. Four Hall linear sensors are installed around alnicos at equal spacings. These sensors collect a magnetic signal and then send it to the MCU. Finally, the absolute angle can be calculated after the MCU is processed.

Figure 2 is a simplified sketch of the magnetic encoder. The dark part is a north pole, and the light part is a south pole, and a north pole and a south pole compose a pole pair. The angle 0° is set as the start point. When the alnico rotates to point A, the absolute angle of A can be calculated via
(1)$$\theta =\left(N_{i}-1\right)\times \frac{360{}^{\circ}}{m}+\frac{\theta_{i}}{m}$$
where
θ is the absolute angle of point A;$N_{i}$ is the pole position, meaning which pair of poles point A is at;$\theta_{i}$ is the angle in a single cycle, which is the angle of point A at the current pole pair;m is the number of pole pairs.

**Figure 2 sensors-21-03095-f002:**
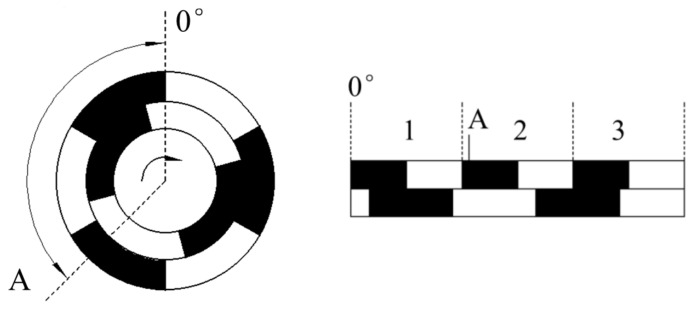
Simplified sketch of the magnetic encoder.

Two linear Hall sensors are installed around each alnico, and the phase angle between them is set as 90°. Then $\theta_{i}$ can be calculated with the arctangent algorithm:(2)$$\theta_{i}=\arctan\frac{V_{A}}{V_{B}}$$
where $V_{A}$ and $V_{B}$ are the voltage signals from Hall sensors A and B, respectively.

The arctangent range is [−90°,90°], so $\theta_{i}$ from Formula (2) cannot be directly used to calculate the absolute angle. Owing to the relative magnitudes of $V_{A}$ and $V_{B}$, the angular range (0°, 360°) can be divided into eight intervals [12]. The angle in each interval can then be calculated via the corresponding formula.

After calculating the angle in a single cycle of $\theta_{i}$, identifying $N_{i}$ is the key to calculate the absolute angle.

## 3. Encoding Principle with Two Multi-Pole Alnicos

Two multi-pole alnicos are used in the encoder. Their positions relative to each other can be used to determine a unique characteristic for each pair of poles on the outer alnico. If each point in each pair of poles on the outer alnico in Figure 2 has a non-redundant corresponding point on the inner alnico, then the inner signal can be used as the unique characteristic to identify which pair of poles the outer signal is from.

However, Figure 3 shows that when the outer alnico has four pairs of poles, and the inner alnico has two, the position relationship between the outer first pole pair and inner alnico is the same as that of the outer third pole pair and inner alnico, so the signal cannot be used to identify a pole pair. Therefore, the uniqueness of the position relationship needs to be proved first.

Consider two random points, *i*1 and *i*2, on the outer alnico and their corresponding points, *j*1 and *j*2, on the inner alnico. The pole positions of these points are $N_{i1}$, $N_{i2}$, $N_{j1}$, and $N_{j2}$. The single-cycle angles of the points are $\theta_{i1}$, $\theta_{i2}$, $\theta_{j1}$, and $\theta_{j2}$ and the absolute angles of *i*1 and *i*2 are $\theta_{1}$ and $\theta_{2}$. The difference in angle between the initial position of the north pole of pole-pair x on the outer alnico and the initial position of the north pole of pole-pair x on the inner alnico is $\theta_{x}$. Figure 4 shows a simplified sketch of the encoder. Using Formula (1), the following equations can be obtained:
(3)$$\left\{\begin{array}{c} \;\theta_{1}=\left(N_{i1}-1\right)\times \frac{360{}^{\circ}}{m}+\frac{\theta_{i1}}{m}=\left(N_{j1}-1\right)\times \frac{360^{\circ }}{n}+\frac{\theta_{j1}}{n}+\theta_{x}\\ \;\theta_{2}=\left(N_{i2}-1\right)\times \frac{360{}^{\circ}}{m}+\frac{\theta_{i2}}{m}=\left(N_{j2}-1\right)\times \frac{360^{\circ }}{n}+\frac{\theta_{j2}}{n}+\theta_{x} \end{array}\right.$$

It can be seen that $\theta_{1}\neq \;\theta_{2}$ when $\theta_{i1}\neq \theta_{i2}$ or $\theta_{j1}\neq \theta_{j2}$. When $N_{i1}$=$N_{i2}$, points *i*1 and *i*2 are in the same cycle, which implies that $\theta_{i1}\neq \theta_{i2}$. Similarly, $\theta_{j1}\neq \theta_{j2}$ when $N_{j1}$=$N_{j2}$. Therefore, the uniqueness of the position relationship can be proved if a condition can be found that satisfies $\theta_{1}\neq \;\theta_{2}$ when $N_{i1}\neq N_{i2}$ and $N_{j1}\neq N_{j2}$.

Assume that $\theta_{i1}=\theta_{i2}$ and $\theta_{j1}=\theta_{j2}$ when $N_{i1}\neq N_{i2}$ and $N_{j1}\neq N_{j2}$. Subtracting one equation in Formula (3) from the other, the following formula can be obtained:(4)$$\frac{\left(N_{i1}-N_{i2}\right)}{\left(N_{j1}-N_{j2}\right)}=\frac{m}{n}$$

Given that $N_{i}\;\leq \;m$, it can be seen that when *m* and *n* are coprime, Formula (4) cannot be satisfied. Therefore, the position relationship is unique when the two alnicos have a coprime number of pole pairs.

Formula (3) can be transformed into the following formula:(5)$$\left\{\begin{array}{c} \;\frac{\theta_{i1}}{m}-\frac{\theta_{j1}}{n}=\left(N_{j1}-1\right)\times \frac{360^{\circ }}{n}-\left(N_{i1}-1\right)\times \frac{360{}^{\circ}}{m}+\theta_{x}\\ \;\frac{\theta_{i2}}{m}-\frac{\theta_{j2}}{n}=\left(N_{j2}-1\right)\times \frac{360^{\circ }}{n}-\left(N_{i2}-1\right)\times \frac{360{}^{\circ}}{m}+\theta_{x} \end{array}\right.$$

Setting $\lambda =\frac{\theta_{i}}{m}-\frac{\theta_{j}}{n}$, it can be seen that for two random points on the outer alnico, $\lambda_{1}\neq \lambda_{2}$ when *m* and *n* are coprime. Because the values of *m*, *n*, and $\theta_{x}$ are defined when the encoder is processed, the amount of λ is limited and can be used as a characteristic value. Therefore, $\frac{\theta_{i}}{m}-\frac{\theta_{j}}{n}$ can be used to calculate λ, λ can be used to determine $N_{i}$, and Formula (1) can be used to calculate θ.

## 4. Calculation of Characteristic Value

When the encoder has been processed, the positions of the outer and inner alnicos relative to each other are determined. Then the characteristic value can be calculated. Figure 5 is the curve of the characteristic value from an encoder with 16 pole pairs on the outer alnico and seven pole pairs on the inner. Using the value to calculate the angle ensures absoluteness. However, in practical work, the corresponding interval of each value has not been determined at this time owing to the unknown installation position error $\theta_{x}$, which means the pole position corresponding to the value cannot be determined.

From Formula (1), the pole position $N_{i}$ can be calculated when the absolute angle θ and single-cycle angle $\theta_{i}$ are known. To improve the accuracy of the encoder, a high-accuracy photoelectric encoder will be used to output the absolute angle for calibration. Then the pole position can be calculated, and the relationship between the pole position and characteristic value can be determined by using pole position to calculate characteristic value.

In practical work, the relationship between the absolute angle and single-cycle angle cannot be strict owing to the error, so the calculated pole position may not be accurate. The algorithm below is proposed to solve the problem.
Use the ideal absolute angle to simulate the output of the photoelectric encoder. Set the first sample point as 0°, the last sample point as 360°, and the step as 0.01°. Then the relationship between the absolute and single-cycle angles of the outer alnico is as shown in Figure 6.Divide the sample points in a whole cycle into M equal intervals. The value of M is based on the number of pole pairs m and number of sample points to make sure M >m and each interval has enough sample points. Figure 6 is a partial curve for M=256.Use the value of the first sample point $\theta_{Mi}$ in interval i to subtract the value of the first sample point $\theta_{Mi-1}$ in interval i−1. Then M−1 differences $D_{i}$ are obtained. Figure 7 shows a partial curve of $D_{i}$ and its corresponding intervals.Set the pole position of the first sample point in the first interval equals to m, then successively judge whether the $D_{i}$ of subsequent intervals is less than 0. If $D_{i}>0$, then the pole position has not changed between the previous and current intervals, so the pole position of the current interval is the same as that of the previous one. If $D_{i}<0$, then the pole position has changed at some point between these two intervals, and the pole position of the current interval is that of the previous one plus 1. When the pole position N is greater than m, set it as 1. In Figure 7, $D_{106}$ represents the situation when $D_{i}>0$, and $D_{109}$ represents $D_{i}<0$. After this processing, the relationship between intervals and pole positions can be determined preliminarily. Figure 8 shows the relationship between partial intervals and pole positions. There is still an error when M=108.Use the pole position of interval i to subtract the pole position of interval i−1 to obtain difference $\tau_{i}$, and the pole position of the first interval to subtract the pole position of interval M to obtain difference $\tau_{1}$. Then M differences are obtained, as shown in Figure 9.Match each difference with corresponding intervals, then find out the intervals with differences of 1 or 1−m; these intervals include sample points when the pole position changes. Combine each of these intervals with its previous one. Then m intervals are obtained. These intervals compose a set T.If the interval of a sample point does not belong to T, there will be no pole position change in the interval. Then the pole position of the point is the same as that of the interval. If the interval of a sample point belongs to T, then the point may be the intersection of pole positions. If the angle of the point is greater than 180°, its pole position is 1 less than that of the interval; if the angle of the point is greater than 180°, then its pole position is the same as that of the interval. After the processing, the corresponding relationship between the single-cycle angle and pole position can be determined. The partial result is shown in Figure 10.

According to Formula (5), $\theta_{x}$ is needed to build the relationship between the characteristic value and pole position. Transforming Formula (5) gives
(6)$$\theta_{x}=\left(N_{2}-1\right)\theta_{n}-\left(N_{1}-1\right)\theta_{m}-\lambda^{*}$$

Calculating the average of the characteristic value $\lambda^{*}$ and putting it into Formula (6) gives the approximate value of $\theta_{x}$. In this simulation, $N_{1}=16$, $N_{2}=7$, and $\theta_{x}=-24.04{}^{\circ}$.

Formula (5) can be written as
(7)$$\lambda =\frac{\theta_{i1}}{m}-\frac{\theta_{i2}}{n}=N_{2}\theta_{n}-N_{1}\theta_{m}+\theta_{x}+\theta_{m}-\theta_{n}$$

If $\theta_{x}$ is brought into Formula (7), then the relationship between the characteristic value and pole position can be built as shown in Table 1.

## 5. Encoder Simulation

The processing flow of the magnetic signal is shown in Figure 11.

We perform a Maxwell simulation of the magnetic signal. The step is 0.01°, anticlockwise is the positive direction, and 36,000 points are taken in the range 0–360° as the raw signal. These points can be thought of as the signal from an alnico rotating clockwise at low speed. The outer alnico has 16 pole pairs, and the inner alnico has 7.

For the simulation, we use an A1326LLHLX-2-T linear Hall sensor from Allegro. The main characteristic parameters of this sensor are shown in Table 2. When the rotor rotates, the linear Hall sensor could turn the change of magnetic field intensity to the change of voltage, then send the voltage signal to MCU to calculate angular. The rotor’s planning speed is 2000 rpm, so this linear Hall sensor’s performance is sufficient.

The position of the Hall sensors is the start point, the sampling points are taken as the abscissa, and the magnetic field intensity is the ordinate. Figure 12 shows the signals of Hall sensors H1 and H2 around the outer alnico, and Figure 13 shows the signals of Hall sensors H3 and H4 around the inner alnico. The solid curves are the signals of H1 and H3, and the dashed curves are those of H2 and H4.

These signals are put into the simulation platform. Figure 14 and Figure 15 show the decoding results from the arctangent algorithm. The outer alnico has 16 groups of angle values from 0° to 360°, and the inner has seven groups of angle values from 0° to 360°, in accord with the setting of the outer and inner alnicos.

Table 1 shows the calculated characteristic value and its relationship with pole position. Then the absolute angle can be calculated through Formula (1). Figure 16 shows the corresponding relationship between sample points and pole positions, and Figure 17 shows the calculated absolute angle. Because the pole position of the first sample point is set at 16, the start point of the absolute angle in Figure 17 is not zero.

## 6. Conclusions

To fulfill the requirements of a motor with a large radial dimension, this paper has presented the theory of a new combined magnetic encoder with two multi-pole alnico rings. First, it was proved that the position relationship between two alnicos is unique when the alnicos each have a coprime number of pole pairs so that the signal of the inner alnico can be used to identify the pole position of the outer alnico. Then a method of calculating the characteristic value was designed, and this value was used to calculate the absolute angle. Finally, the encoding and decoding theory was verified through simulation.

## Figures and Tables

**Figure 1 sensors-21-03095-f001:**
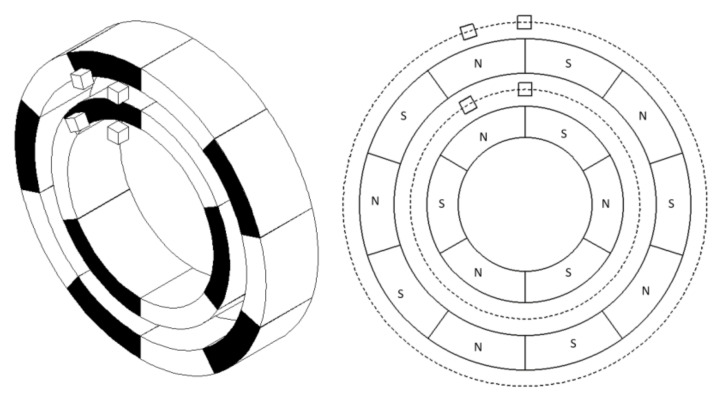
Schematic of the magnetic encoder’s signal generation principle.

**Figure 3 sensors-21-03095-f003:**
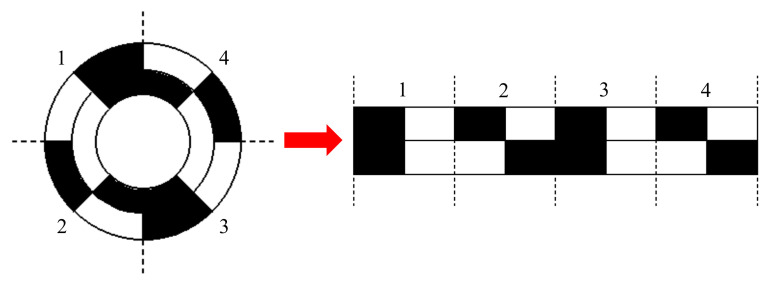
Alnicos with the same position relationship.

**Figure 4 sensors-21-03095-f004:**
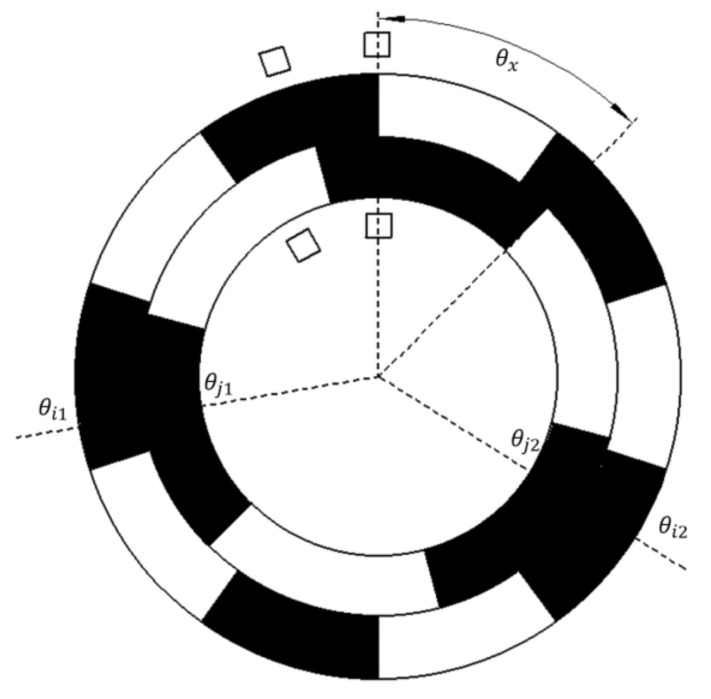
Simplified sketch of the magnetic encoder with two random points.

**Figure 5 sensors-21-03095-f005:**
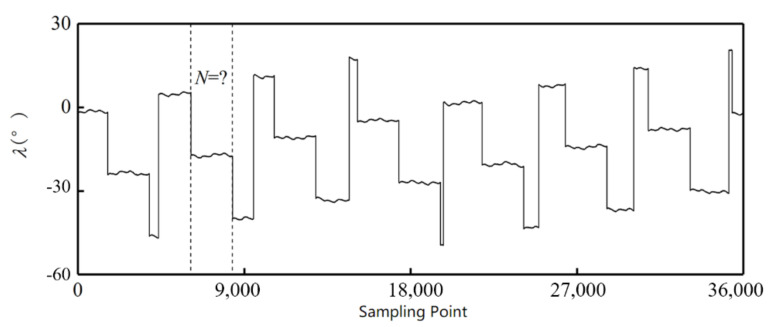
Calculated characteristic value.

**Figure 6 sensors-21-03095-f006:**
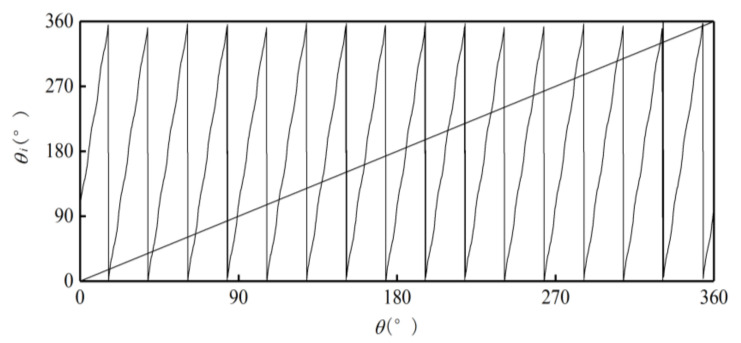
Relationship between ideal angle and single-cycle angle of outer alnico.

**Figure 7 sensors-21-03095-f007:**
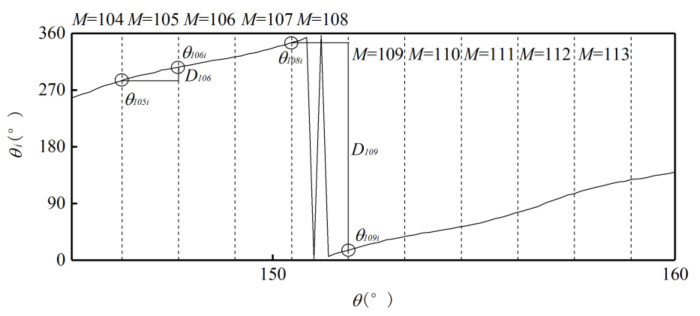
Partial curve with intervals.

**Figure 8 sensors-21-03095-f008:**
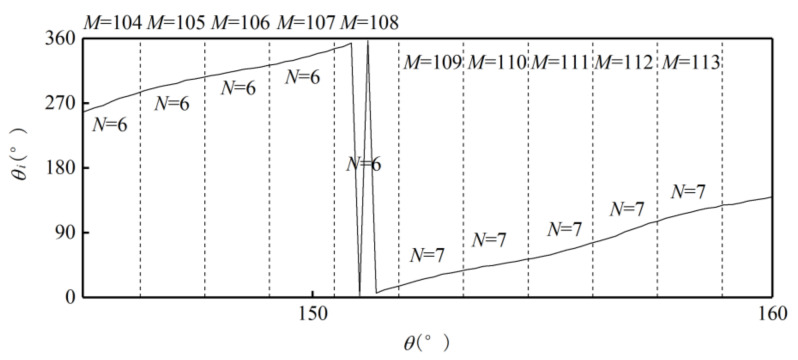
Primary corresponding relationship between intervals and pole positions.

**Figure 9 sensors-21-03095-f009:**
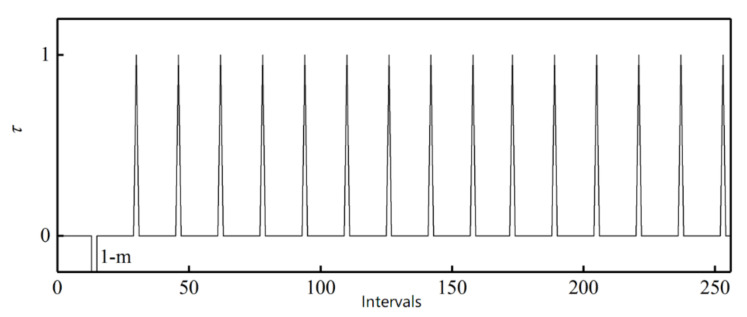
Differences of all intervals.

**Figure 10 sensors-21-03095-f010:**
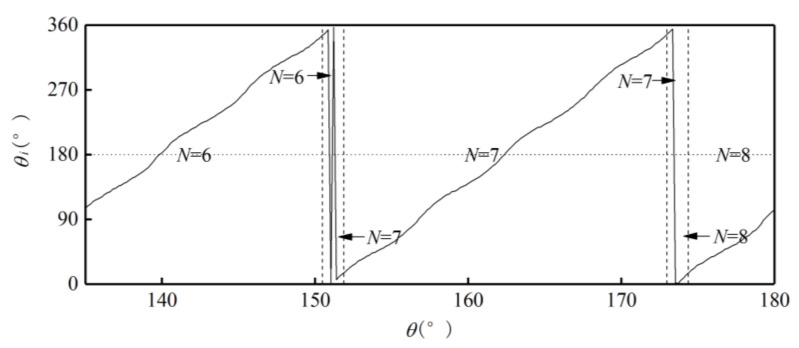
Relationship between single-cycle angle and pole position.

**Figure 11 sensors-21-03095-f011:**
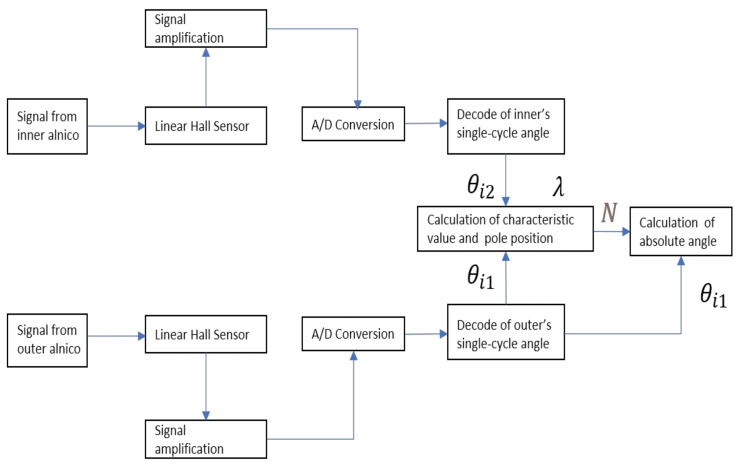
Processing of absolute angle calculation based on characteristic value.

**Figure 12 sensors-21-03095-f012:**
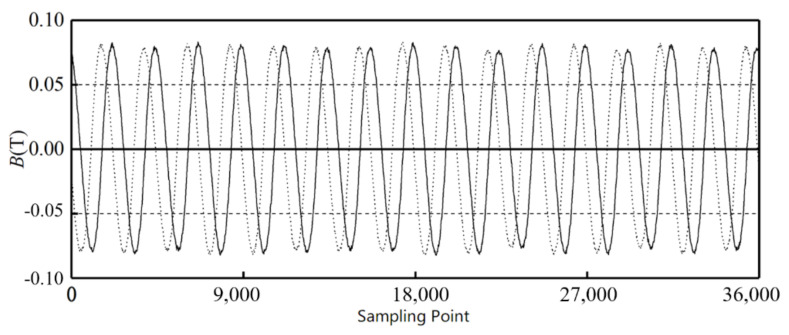
Signals from Hall sensors of outer alnico.

**Figure 13 sensors-21-03095-f013:**
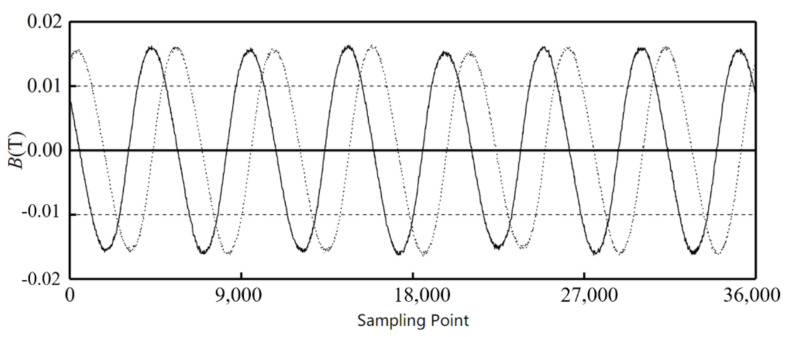
Signals from Hall sensors of inner alnico.

**Figure 14 sensors-21-03095-f014:**
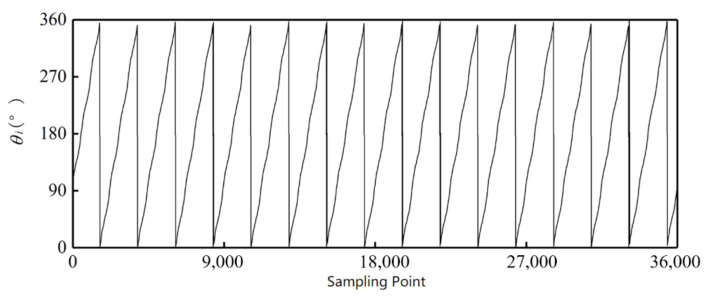
Decoding result for outer alnico.

**Figure 15 sensors-21-03095-f015:**
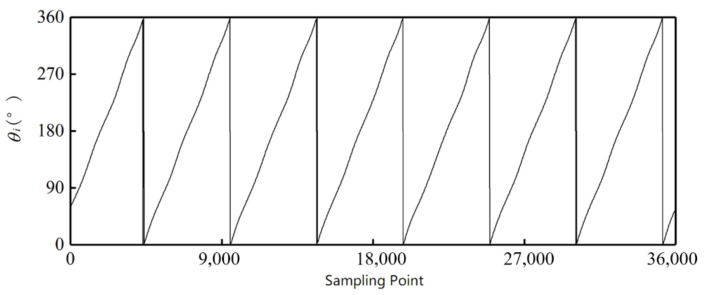
Decoding result for inner alnico.

**Figure 16 sensors-21-03095-f016:**
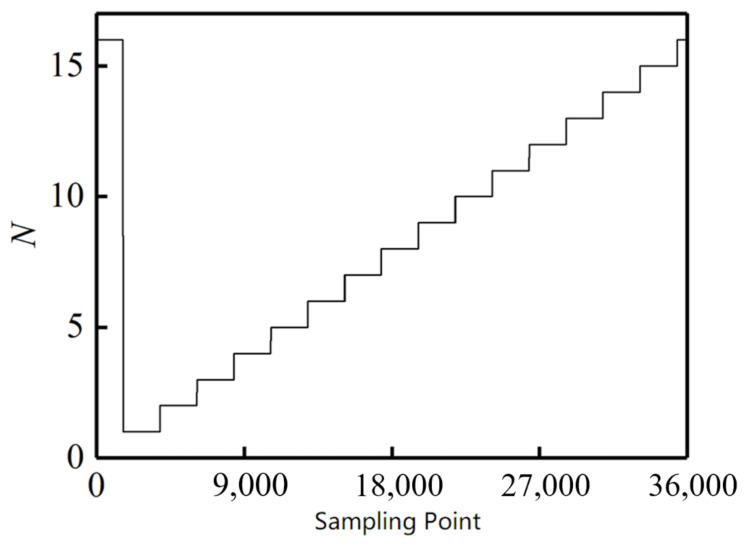
Decoding result.

**Figure 17 sensors-21-03095-f017:**
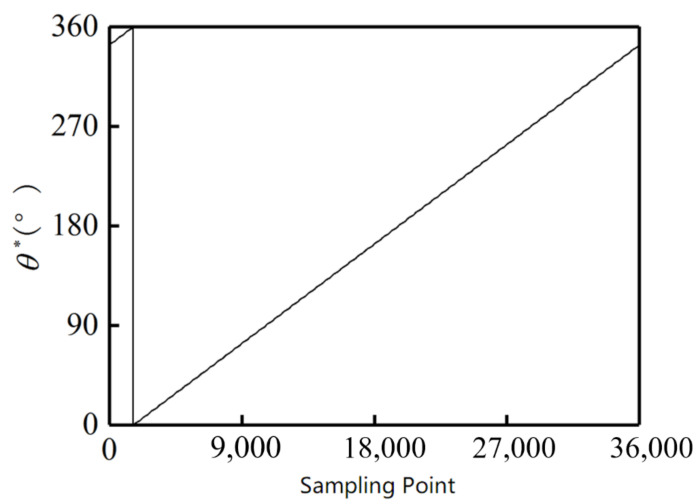
Calculated angle.

**Table 1 sensors-21-03095-t001:** Characteristic values and corresponding outer alnico pole positions.

Pole Position *N*	Characteristic Value *λ*	Pole Position *N*	Characteristic Value *λ*	Pole Position *N*	Characteristic Value *λ*
1	−24.04	6	17.7457	12	−14.3971
2	−46.54	7	−4.7543	13	−36.8971
2	4.8886	8	−27.3543	13	14.5314
3	−17.6114	9	−49.7543	14	−7.9686
4	−40.1114	9	1.6743	15	−30.4686
4	11.3171	10	−20.8257	15	20.6900
5	−11.1829	11	−43.3257	16	−1.54
6	−33.6829	11	8.1029		

**Table 2 sensors-21-03095-t002:** The main characteristic parameters of the linear Hall sensor.

Model Number	Rated Working Voltage (V)	Internal Bandwidth (kHz)	Magnetic Sensitivity Coefficient (mV/G)	Output Referred Noise (mV)
A1326LLHLX-2-T	5	17	4.750~5.250	3.5

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
