# Peer review of "A New Kind of Absolute Magnetic Encoder"

_sensors, 2021, doi:10.3390/s21093095_

Round 1

Reviewer 1 Report

Dear author, 

Thank you for your publication, which present an interresting way to construct an absolute sensor using 2 magnectic encoders and no 1-pulse-per-rev-tachometer.

From my point of view, major modifications must be made before this paper is published. and before presenting those, I want to mention one subject that would definitely prevent this paper to be published:

  • citation [12] is very similar to the present paper (same authors, same topic). Unfortunately, I can only say that reading its title and abstract: I have no access to its content. You must clearly explain what has been done in this old paper and what is new in the present paper.  

Now let me propose the major modification remarks: 

  1. Introduction is very weak. there is no real state of the art. just to explain my point: every paper sourced in the refrence have been writtend after 2016... Although the subject of proposing an absolute magnetic encoder is clearly not new.
  2. Throughout the paper, the notation is very hard to grasp. The demonstration in section 3 it too long. Maybe ask for advice to someone who could help you formalizing more consisely. The coprime number condition is quite simple to grasp (or I missed something in the theorem you proove: maybe should you more precisely state what you want to prove). It seems you look for the properties of the residu of the euclidian division of two coprime numbers.
  3. Same remark for section 4, the notation makes the reading very hard to understand, though the problem seems quite simple.
  4. english is hard to read. many sentences cannot be understood. 

Reviewer 2 Report

In this manuscript, the authors described a new kind of absolute magnetic encoder. In general, the idea sounds interesting. However, it is missing the physical properties of the four Hall sensors. What is the time response? What is the level of the noise? Which current you use in these Hall sensors? What is the speed of the rotor rotating? The answer to these questions will clarify the practical work of your encoder

Round 2

Reviewer 2 Report

After a significant improvement of the revised manuscript, I believe that this paper is ready for publication